# High-resolution ultrasonography for early diagnosis of neural impairment in seropositive leprosy household contacts

**Andrea De Martino Luppi**[ID][1,2,3☯], **Guilherme Emilio Ferreira**[2☯], **Denis Luiz Prudêncio**[1☯], **Douglas Eulálio Antunes**[1,3☯], **Lúcio Araújo**[4☯], **Diogo Fernandes dos Santos**[1,3☯], **Marcello Henrique Nogueira-Barbosa**[5☯], **Isabela Maria Bernardes Goulart**[ID][1,3☯]*

1 National Reference Center for Sanitary Dermatology and Leprosy, Clinics' Hospital, Federal University of Uberlândia (UFU/EBSERH), Uberlândia, MG, Brazil, 2 Radiology Division, Clinics' Hospital, Federal University of Uberlândia (UFU/EBSERH), Uberlândia, MG, Brazil, 3 Postgraduate Program in Health Sciences, School of Medicine, Federal University of Uberlândia (UFU), Uberlândia, MG, Brazil, 4 Department of Mathematics, Federal University of Uberlândia (UFU), Uberlândia, MG, Brazil, 5 Department of Medical Imaging, Hematology and Clinical Oncology, Radiology Division, Ribeirão Preto Medical School, University of São Paulo Ribeirão Preto, São Paulo, Brazil

☯ These authors contributed equally to this work.
* imbgoulart@gmail.com

**Data Availability Statement:** All relevant data are within the manuscript and its Supporting Information files.

## Abstract

Leprosy household contacts (HC) represent a high-risk group for the development of the disease. Anti-PGL-I IgM seropositivity also increases the risk of illness. Despite significant advances in leprosy control, it remains a public health problem; and early diagnosis of this peripheral neuropathy represents one of the main goals of leprosy programs. The present study was performed to identify neural impairment in leprosy HC by analyzing differences in high-resolution ultrasonographic (US) measurements of peripheral nerves between leprosy HC and healthy volunteers (HV). Seventy-nine seropositive household contacts (SPHC) and 30 seronegative household contacts (SNHC) underwent dermato-neurological examination and molecular analysis, followed by high-resolution US evaluation of cross-sectional areas (CSAs) of the median, ulnar, common fibular and tibial nerves. In addition, 53 HV underwent similar US measurements. The US evaluation detected neural thickening in 26.5% (13/49) of the SPHC and only in 3.3% (1/30) among the SNHC (p = 0.0038). The CSA values of the common fibular and tibial nerves were significantly higher in SPHC. This group also had significantly greater asymmetry in the common fibular and tibial nerves (proximal to the tunnel). SPHC presented a 10.5-fold higher chance of neural impairment (p = 0.0311). On the contrary, the presence of at least one scar from the BCG vaccine conferred 5.2-fold greater protection against neural involvement detected by US (p = 0.0184). Our findings demonstrated a higher prevalence of neural thickening in SPHC and support the role of high-resolution US in the early diagnosis of leprosy neuropathy. The combination of positive anti-PGL-I serology and absence of a BCG scar can identify individuals with greater chances of developing leprosy neuropathy, who should be referred for US examination, reinforcing the importance of including serological and imaging methods in the epidemiological surveillance of leprosy HC.

**Funding:** The author(s) received no specific funding for this work.

**Competing interests:** The authors have declared that no competing interests exist.

## Introduction

Leprosy is an infectious disease caused by *Mycobacterium leprae (M. leprae)*, an obligate intra-cellular parasite with a predilection to infect peripheral nerves and the skin. It represents the main etiology of non-traumatic peripheral neuropathy worldwide, while Brazil ranks second globally in the number of new cases [1,2].

Leprosy still represents a challenge for public health policies, since it continues to be the main cause of disability among infectious diseases. This is mainly a reflection of the delay between the onset of symptoms and diagnosis, which can last for years, during which the disease evolves slowly and progressively [3,4]. The predominance of multibacillary cases with neural impairment suggests failure in early diagnosis and indicates continuous transmission of the disease [5,6].

Leprosy household contacts (HC) of multibacillary patients have a 5 to 10 times greater risk of illness than the general population. In addition, the possibility of dissemination of the bacillus may be related to healthy carriers and individuals with subclinical infections, defined by a positive serological test in Enzyme-linked immunosorbent assay (ELISA) anti-phenolic glyco-lipid I (anti-PGL-I) Immunoglobulin M (IgM) [7–10].

Therefore, it is important to emphasize the importance of investigating subclinical neural involvement in this group, which is still poorly understood in the literature. Such an investigation would enable the discussion on chemoprophylaxis and early treatment as a strategy for leprosy control [8,11–13].

Recently, the high-resolution US technique has emerged as a new tool for the assessment of peripheral neuropathy, allowing the study of nerve morphology along its course. The most important sonographic parameter is the cross-sectional area (CSA), which aims to assess the presence of nerve enlargement, observed in several types of polyneuropathies, such as inflammatory, hereditary and also infectious neuropathies [14–19]. Neural thickening can be identified clinically by palpation, but this method is subjective and has low accuracy [17,20–22]. Although the literature indicates to the significance of high-resolution US in leprosy patients [14–17,21], there are few studies available on the evaluation of peripheral nerve impairment by this technique encompassing leprosy HC [23].

Our study was carried out to identify subclinical neural impairment in leprosy HC by analyzing differences in US measurements of peripheral nerves between leprosy contacts (seropositive and seronegative) and healthy volunteers (HV).

## Materials and methods

### Ethics statement

We recruited leprosy HC from the National Reference Center of Sanitary Dermatology and Leprosy in Brazil, under the approval of the Ethics Committee of the Federal University of Uberlândia (CAAE: 23136419.3.0000.5152). Written informed consent was given by all adult participants, and was obtained from parents of participating minors on their behalf.

### Type of study and subjects

This is a cross-sectional study composed of 3 groups, encompassing 49 seropositive HC (SPHC), 30 seronegative HC (SNHC) and 53 HV, enrolled by intentional sampling at the National Reference Center for Sanitary Dermatology and Leprosy–Clinical Hospital, Medical School, Federal University of Uberlândia, from September 2019 to March 2021. From this period, 805 HC were notified. A proportion of 70.1% (564/805) of these attended the initial evaluation, when all were submitted to anti-PGL-I serology collection. Among them, 22%

(124/564) were seropositive. In this study, 39.5% (49/124) SPHC were submitted to all complementary tests at the time when seropositivity to the ELISA anti-PGL-I IgM was confirmed. It is important to point out that, despite having a significant sample, the SARS-COV2 pandemic made it difficult for patients to access the health service and adhere to it in a first evaluation.

As eligibility criteria, for the group of SPHC, the participants presented a history of domiciliary contact with a case of leprosy and positivity in the ELISA anti-PGL-I IgM. For the group of SNHC, the participants had a history of domiciliary contact with a case of leprosy and negativity in the ELISA anti-PGL-I IgM. The group of HV was composed of participants of the same population from an endemic region for leprosy, but without a history of domiciliary contact with cases of leprosy. It is important to highlight that the SNHC were also healthy subjects, and had no other plausible cause for peripheral neuropathy.

Participants who presented other possible etiologies of peripheral neuropathies, such as diabetes mellitus, hypothyroidism, hepatitis B or C, human immunodeficiency virus infection, hereditary neuropathies or chronic alcoholism were excluded from both groups.

## Clinical characterization

In Brazil, the epidemiological investigation of leprosy HC consists of anamnesis, dermato-neurological examination and vaccination with BCG for contacts without signs and symptoms of leprosy at the time of evaluation. A new dose of BCG is applied to contacts with none or only one vaccine scar [24]. At this center, leprosy HC are followed up for a period of at least 7 years, annually, when they are submitted to serological analysis. They are classified into intra- or extradomiciliary, and also into contact with pauci- or multibacillary, according to the operational classification of the index case.

During dermato-neurological examination, each contact was carefully inspected not only for skin lesions, but also for neurological impairment, through the presence of sensory and motor impairment. All HC underwent a rigorous sensory evaluation, ruling out the impairment of all sensory modalities (pain, temperature, touch and vibration). Furthermore, the contact groups underwent clinical palpation of nerves, always by the same trained professional, to detect peripheral nerve thickening: ulnar nerve at the elbow, common fibular at the fibular head and tibial nerve at the ankle. Clinical palpation of the median nerve could not be performed due to its deeper location. It is noteworthy that all HV were submitted to dermato-neurological examination to rule out leprosy, even in the absence of epidemiological antecedents.

The epidemiological characteristics were also evaluated: age, gender, type of contact (intra-domiciliary or extradomiciliary), operational classification of the index case (paucibacillary or multibacillary) and the absence or presence of at least one scar of BCG vaccine.

## Laboratory analyzes

All SPHC and SNHC underwent the following laboratory tests, as described below.

Identification of acid-fast bacilli (AFB)–This analysis was performed on slit-skin smears from six sites (two ear lobes, two elbows, two knees), as well as on skin biopsy samples. Sample collection was preceded by topical application of cream containing lidocaine (7%) and tetracaine (7%) at all sites.

ELISA anti-PGL-I IgM serology–Serum anti-PGL-I IgM antibodies were detected by ELISA performed against the purified native PGL-I from the *M. leprae* cell wall. The reagent was obtained through BEI Resources, NIAID, NIH: Monoclonal Anti-*Mycobacterium leprae* PGL-I, Clone CS-48 (produced in vitro), NR-19370. The titration of anti-PGL-I antibodies was expressed as an ELISA index according to the proportion between the bacillary load of the sample in relation to the cutoff point. Values above 1.0 were considered positive [25].

DNA Extraction and Real Time Quantitative Polymerase Chain Reaction (qPCR) of the following samples: 1- slit-skin smear (one sample) from six sites (two ear lobes, two elbows, two knees); 2- elbow skin biopsy. The qPCR assay to detect *M. leprae* DNA was performed by targeting the bacillus-specific genomic region (RLEP) in a real-time PCR system (ABI 7300, Applied Biosystems, Foster City, CA, USA) [26–28].

## Skin biopsy

All of the leprosy HC selected did not present any skin lesion. For this reason, the biopsy of a small elbow skin fragment was performed, considering that it is a cold region with possible intradermal neural impairment and, therefore, a site often altered in leprosy neuropathy. A wedge-shaped incision was made using a scalpel blade, and a fragment of approximately 1 cm along its greatest length that reached the hypodermis was removed. One part of the skin sample was sent to the molecular pathology and biotechnology laboratory and the other part was sent to the pathology laboratory for histopathological evaluation. Fite-Faraco staining was used to investigate *M. leprae*.

## Ultrasonography

All of the high-resolution US sessions were performed on leprosy HC and HV by a Board-certified radiologist, with experience in peripheral nerve imaging, using a linear transducer model Esaote MyLab™50 XVision (ESAOTE LATINOAMERICA, Huixquilucan, Mexico) at a broadband frequency of 6–18 MHz. In order to avoid verification bias, the investigator was blinded regarding serological results of HC, preventing interference in US outcomes.

Participants were examined in a seated position with the arm in abduction and elbow flexed at 45˚ for assessment of ulnar and knees flexed at 90˚ for the common fibular nerves. For median nerve examination, the arms of the study participant were positioned by their respective sides and in supination. The tibial nerve was examined in minimal external rotation of the lower limb. Positioning of limbs of study participants during US were kept uniform throughout the study.

US measurements were performed at compression sites often affected in leprosy neuropathy. The ulnar nerve was evaluated at the ulnar sulcus in the cubital tunnel (Ut) and at 3 to 4 cm above the medial epicondyle, proximal to the cubital tunnel (Upt) [14,15]. The median nerve was scanned at the wrist in the carpal tunnel (Mt) and 3 to 4 cm above the carpal tunnel (Mpt). The common fibular nerve was evaluated at the level of the fibular head. The tibial nerve was scanned at the ankle in the tarsal tunnel (Tt), behind the medial malleolus, and at 3 to 4 cm above the medial malleolus, proximal to the tarsal tunnel (Tpt) [23].

For measuring cross-sectional areas (CSAs) of the nerves, the US beam was kept perpendicular to the nerve to minimize anisotropy. CSAs were measured by freehand delimitation at the inner borders of the echogenic rims of the nerves, using the electronic cursor at the time of examination [29].

CSA measurements were utilized to determine the CSA index (ΔCSA), which was calculated as the absolute difference between CSAs for each nerve point from one side to the contralateral side. High ΔCSA values reflect nerve asymmetry with the contralateral nerve [29].

Moreover, we also calculated the absolute difference between CSAs measurements of each nerve in the tunnel and proximal to the tunnel points (Δtpt): Mt-Mpt index (ΔMtpt) of the median nerves, Ut-Upt index (ΔUtpt) of the ulnar nerves, and Tt-Tpt index (ΔTtpt) of the tibial nerves. High ΔMtpt, ΔUtpt and ΔTtpt values reflect non-uniform enlargement of the nerves [20,29].

In summary, as outcome factors, we assessed the following variables: CSA and ΔCSA of each peripheral nerve, and ΔMtpt, ΔUtpt and ΔTtpt.

The HV group was evaluated in detail by a radiologist with experience in the evaluation of peripheral nerves and, although there was no pairing specifically for this study, the data obtained in this group are in accordance with the normality standards used in other studies [22,30–32]. Considering that the US assessment of peripheral nerves is still a recent technique, we believe that internal standardization is important. It is also important to highlight that the US data of this control group are used in the radiology laboratory of this service, including for comparison with other neuropathies, such as those of inflammatory and hereditary etiology.

For the classification of the values of CSA, ΔCSA, ΔMtpt, ΔUtpt and ΔTtpt as normal or abnormal, the measurements obtained in the evaluation of the nerves of HV were used, considering any values greater than mean plus 3 standard deviations as abnormal.

## Statistical analysis

The Shapiro Wilk test was employed to test data normality within groups. As all ultrasound variables did not present normal probability distribution, we performed the Kruskal-Wallis test to analyze differences among the means of the three groups. The Chi-square test was applied for the study of dichotomous variables. For continuous variables, the Mann-Whitney u test was used. Simple and multiple logistic regression were utilized to verify the dependence relation between the presence of US abnormality (categorical variable) and the independent variables (ELISA anti-PGL-I IgM, slit-skin smear qPCR, skin biopsy qPCR and BCG scar), followed by the selection of variables by the *stepwise* method. Probability ($p$) values less than 0.05 were considered significant. The procedures were performed using the software Statistical Package for Social Sciences—SPSS Version 20 (IBM, Armonk, NY, USA) for Windows.

## Results

In our study, the total sample size was 132 participants, subdivided into 3 groups: 49 SPHC, 30 SNHC and 53 HV, according to the eligibility criteria. The HV group was composed of 33 women and 20 men, with a mean age of 40.9 ± 12.0 years, without statistical differences in relation to HC. For clinical data and CSAs values of each participant see supporting information (S1–S3 Tables).

Comparisons of epidemiological and clinical characteristics between SPHC and SNHC did not differ significantly, as presented in Table 1. The presence of neural thickening was clinically observed in only 2.0% (1/49) of SPHC and none of the SNHC. None of the evaluated HC presented skin lesion, sensory symptoms or muscular weakness. The mean ELISA anti-PGL-I IgM index was 2.02 in SPHC and 0.37 in SNHC ($p<0.0001$). The positivity of skin biopsy and slit-skin smear qPCR analysis did not show any significant difference between these groups (Table 1). The bacilloscopic tests of the slit-skin smear and the skin biopsy were negative in all HC, as well as the histopathological evaluation of the skin biopsy.

A total of 392 nerves (ninety-eight each of ulnar, median, common fibular and tibial nerves) were assessed in SPHC, 240 nerves (60 nerves of each) in SNHC and 424 nerves (106 nerves of each) in 53 HV. We excluded one measurement of the common fibular nerve from a subject of the SPHC group due to a previous history of fibula fracture, which may be related to damage to the respective nerve in that location [33]. In addition, eleven measurements of the Mt nerve were excluded (eight measurements from SPHC and three from SNHC group) due to ultrasonographic evidence of carpal tunnel syndrome (measurement at Mt corresponded to more than twice the measurement of the same nerve at the Mpt) [34]. In our study, we investigated

**Table 1. Epidemiological, clinical and laboratory characteristics among leprosy household contacts.**

| Parameters | Seropositive HC (n = 49) | Seronegative HC (n = 30) | *p* value |
|---|---|---|---|
| **Age** | 42.57 ± 16.30 | 42.30 ± 15.36 | 0.9476 |
| **Gender** | | | 0.1124 |
| Male | 20.4% (10/49) | 36.7% (11/30) | |
| Female | 79.6% (39/49) | 63.3% (19/30) | |
| **Type of contact** | | | 0.2595 |
| Intradomiciliary | 53.1% (26/49) | 40% (12/30) | |
| Extradomiciliary | 46.9% (23/49) | 60% (18/30) | |
| **BCG** | | | 0.6201 |
| 0 scars | 20.4% (10/49) | 13.3% (4/30) | |
| 1 or 2 scars | 79.6% (39/49) | 86.7% (26/30) | |
| **Index case** | | | 0.9232 |
| Paucibacillary | 6.1% (3/49) | 6.7% (2/30) | |
| Multibacillary | 93.9% (46/49) | 93.3% (28/30) | |
| **Clinical evaluation** | | | |
| Neural thickening | 2.0% (1/49) | 0 | 0.3264 |
| Skin lesions | 0 | 0 | - |
| Sensory symptoms | 0 | 0 | - |
| Muscular weakness | 0 | 0 | - |
| **Laboratory analyzes** | | | |
| ELISA index | 2.02 ± 0.73 | 0.37 ± 0.21 | 0.0001* |
| Positivity skin biopsy qPCR | 6.1% (3/49) | 3.3% (1/30) | 0.5723 |
| Positivity slit-skin qPCR | 12.2% (6/49) | 6.6% (2/30) | 0.4126 |
| Abnormal US | 26.5% (13/49) | 3.3% (1/30) | 0.0038* |

HC = household contacts; n = number of contacts; BCG = Bacillus Calmette-Guérin; ELISA = enzyme-linked immunosorbent assay; qPCR = Real Time Quantitative Polymerase Chain Reaction; US = Ultrasonography.

*Statistically significant.

all participants for the presence of Doppler signal; however, none of them showed signs of intra and perineural hyperemia on Doppler.

US evaluation detected neural enlargement in 26.5% (13/49) of SPHC and in only one SNHC (*p* = 0.0038) (Table 1). Among SPHC with thickening detected by US, the mean number of nerves affected was 1.8 per contact, while 53.8% (7/13) presented only one altered nerve (mononeuropathy) and 46.2% (6/13) two or more altered nerves (multiple mononeuropathy). The nerves most frequently affected were the common fibular (Fig 1) and the tibial (Fig 2) nerves, as described in Table 2.

The following tables show the measurements obtained for CSA, ΔCSA, ΔMtpt, ΔUtpt and ΔTtpt of the nerves assessed on HV, SPHC and SNHC.

The mean values of CSA of the common fibular, tibial at the tunnel (Tt) and tibial proximal to the tunnel (Tpt) nerves were significantly higher in SPHC compared to the other groups, as described in Table 3. The other nerves did not present any significant difference between the groups.

SPHC showed significantly higher ΔCSAs compared to SNHC at the common fibular and tibial proximal to the tunnel (Tpt), reflecting asymmetry with the contralateral side. We observed that SNHC presented significantly lower ΔCSAs compared to HV for the common fibular nerve only. The other nerves did not present any significant difference between the groups (Table 4).

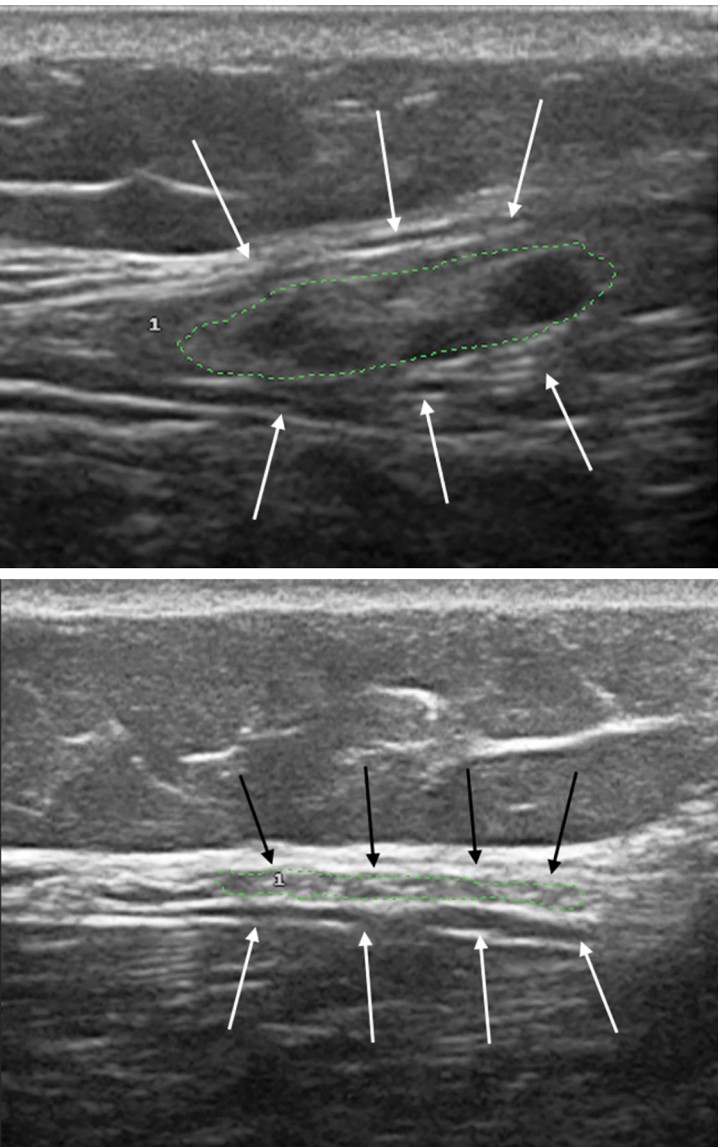

**Fig 1.** High-resolution ultrasonography of common fibular nerve (transverse view) in a seropositive contact (A), showing neural thickening, compared to normal nerve in a seronegative contact (B).

Regarding the absolute difference in measurements between the two points of the nerve (Table 5), none of the nerves showed a statistically significant difference between the groups.

Multiple logistic regression was conducted to verify the dependence relation between the independent variables (ELISA anti-PGL-I IgM, slit-skin smear qPCR, skin biopsy qPCR and BCG scar), and the chance of occurrence of US neural thickening (categorical variable). It was demonstrated that ELISA anti-PGL-I IgM positivity confers a 10.5-fold greater chance of neural damage (OR = 10.48; 95% CI: 1.24 to 88.61; $p$ = 0.0311). The presence of at least one BCG vaccine scar demonstrated 5.2-fold greater protection against neural impairment (OR = 0.19; 95% CI: 0.05–0.75; $p$ = 0.0184). There was no dependence relation with the variables slit-skin smear and skin biopsy qPCR (Table 6). As the OR is less than 1.0, it is a protection factor. Therefore, the best way to find the magnitude of this protection factor is by calculating:

Protection factor = 1/OR = 1.0/0.19 = 5.2

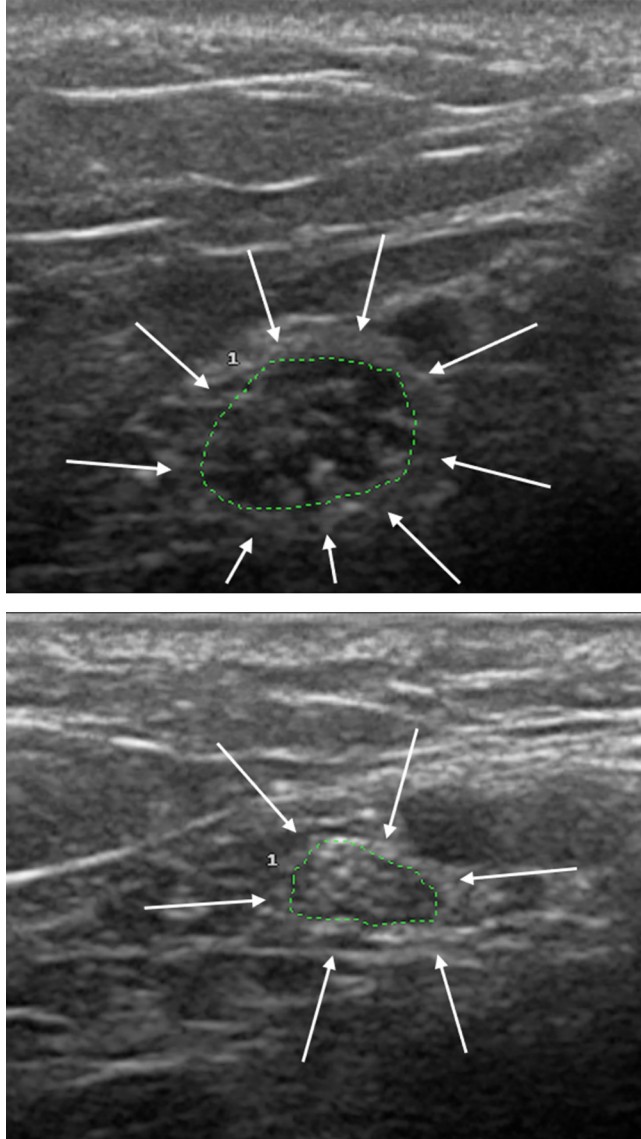

**Fig 2.** High-resolution ultrasonography of tibial nerve (transverse view) in a seropositive contact (A), showing neural thickening, compared to normal nerve in a seronegative contact (B).

**Table 2. Distribution of peripheral nerves most involved in high-resolution ultrasonography evaluation of seropositive contacts.**

| Affected nerves | n | % |
|---|---|---|
| Median | 0 | 0 |
| Ulnar | 2 | 8.7% |
| Tibial | 9 | 39.1% |
| Common fibular | 12 | 52.2% |
| Total of affected nerves | 23 | 100% |

n = number of affected nerves; % = percentage of affected nerves.

**Table 3. Cross-sectional area (CSA) measurements (mm$^2$) by high-resolution ultrasonography in healthy volunteers and leprosy household contacts.**

| Nerves | | Healthy volunteers | Seropositive HC | Seronegative HC | *p1 value* | *p2 value* | *p3 value* |
|---|---|---|---|---|---|---|---|
| **Median (Mpt)** | n | 106 | 98 | 60 | | | |
| | Mean ± SD | 5.97 ± 1.12 | 5.85 ± 1.14 | 5.97 ± 1.33 | >0.9999 | >0.9999 | >0.9999 |
| | Median (Q$_1$/Q$_3$) | 6 (5/7) | 6 (5/7) | 6 (5/7) | | | |
| | 95% CI | [5.69–6.25] | [5.56–6.17] | [5.49–6.45] | | | |
| | Abnormal (>9.33) | 1 (0.94%) | 0 | 1 (1.66%) | | | |
| **Median (Mt)** | n | 106 | 90 | 57 | | | |
| | Mean ± SD | 7.62 ± 1.35 | 7,18 ± 1.37 | 7,42 ± 1.28 | 0.2174 | >0.9999 | 0.4599 |
| | Median (Q$_1$/Q$_3$) | 7 (7/8) | 7 (6/8) | 7 (6.75/8) | | | |
| | 95% CI | [7.28–7.97] | [6.83–7.58] | [6.99–7.98] | | | |
| | Abnormal (>11.67) | 0 | 0 | 0 | | | |
| **Ulnar (Upt)** | n | 106 | 98 | 60 | | | |
| | Mean ± SD | 4.81 ± 1.09 | 5.0 ± 1.36 | 4.56 ± 0.96 | >0.9999 | 0.8035 | 0.6542 |
| | Median (Q$_1$/Q$_3$) | 5 (4/6) | 5 (4/5) | 5 (4/5) | | | |
| | 95% CI | [4.53–5.09] | [4.63–5.37] | [4.23–4.9] | | | |
| | Abnormal (>8.08) | 0 | 2 (2.04%) | 0 | | | |
| **Ulnar (Ut)** | n | 106 | 98 | 60 | | | |
| | Mean ± SD | 5.81 ± 1.24 | 5.88 ± 1.48 | 5.55 ± 1.14 | >0.9999 | >0.9999 | >0.9999 |
| | Median (Q$_1$/Q$_3$) | 6 (5/7) | 6 (5/6) | 6 (5/6) | | | |
| | 95% CI | [5.5–6.12] | [5.46–6.28] | [5.15–5.95] | | | |
| | Abnormal (>9.53) | 0 | 2 (2.04%) | 0 | | | |
| **Common Fibular** | n | 106 | 97 | 60 | | | |
| | Mean ± SD | 12,08 ± 2,15 | 14,14 ± 4,21 | 11,7 ± 1,97 | 0.0025* | 0.8149 | 0.0002* |
| | Median (Q$_1$/Q$_3$) | 12 (10/14) | 13 (11.75/16) | 12 (10/13) | | | |
| | 95% CI | [11.5–12.65] | [13.11–15.36] | [10.97–12.43] | | | |
| | Abnormal (>18.53) | 0 | 12 (12.37%) | 0 | | | |
| **Tibial (Tpt)** | n | 106 | 98 | 60 | | | |
| | Mean ± SD | 8,47 ± 1,19 | 9,82 ± 2,66 | 8,6 ± 1,49 | 0.0003* | >0.9999 | 0.0212* |
| | Median (Q$_1$/Q$_3$) | 8 (8/9) | 9 (8/11) | 9 (8/10) | | | |
| | 95% CI | [8.18–8.77] | [9.11–10.57] | [8.05–9.15] | | | |
| | Abnormal (>12.04) | 0 | 9 (9.18%) | 0 | | | |
| **Tibial (Tt)** | n | 106 | 98 | 60 | | | |
| | Mean ± SD | 8,47 ± 1,26 | 9,77 ± 2,51 | 8,63 ± 1,54 | 0.0003* | >0.9999 | 0.0484* |
| | Median (Q$_1$/Q$_3$) | 8 (8/10) | 9 (8/11) | 9 (8/10) | | | |
| | 95% CI | [8.17–8.78] | [9.06–10.45] | [8.09–9.18] | | | |
| | Abnormal (>12.25) | 0 | 8 (8.16%) | 0 | | | |

HC = household contacts; n = number of nerves; SD = standard deviation; Q$_1$ = 25th percentile; Q$_3$ = 75th percentile; 95% CI– 95% confidence interval;

Abnormal = number of nerves with abnormal measurements and percentages in parentheses; *p1 value* = healthy volunteers vs. seropositive HC; *p2 value* = healthy volunteers vs. seronegative HC; *p3 value* = seropositive HC vs. seronegative HC.

*Statistically significant.

## Discussion

This study aimed to describe neural ***thickening*** in leprosy HC through US assessment. Previous researches have documented neural involvement in leprosy contacts [13,35,36], but few studies explored the possible morphological abnormalities of nerves on US study in this population [23]. Although several studies have shown the importance of US evaluation for the diagnosis of neuropathy in leprosy patients [14,16–22,30,37,38], few have evaluated the possible neural damage by US assessment in leprosy HC [23].

**Table 4. Absolute difference in cross-sectional area (ΔCSA) measurements (mm$^2$) between right and left sides by high-resolution ultrasonography in healthy volunteers and leprosy household contacts.**

| Nerves | | Healthy volunteers | Seropositive HC | Seronegative HC | *p1 value* | *p2 value* | *p3 value* |
|---|---|---|---|---|---|---|---|
| **Median (Mpt)** | n | 53 | 49 | 30 | | | |
| | Mean ± SD | 0.66 ± 0.71 | 0.63 ± 0.64 | 0.47 ± 0.57 | >0.9999 | 0.6917 | 0.7506 |
| | Median (Q$_1$/Q$_3$) | 1 (0/1) | 1 (0/1) | 0 (0/1) | | | |
| | 95% CI | [0.47–0.85] | [0.45–0.82] | [0.25–0.68] | | | |
| | Abnormal (>2.79) | 0 | 0 | 0 | | | |
| **Median (Mt)** | n | 53 | 44 | 28 | | | |
| | Mean ± SD | 0.72 ± 0.79 | 0.75 ± 0.81 | 0.57 ± 0.57 | >0.9999 | >0.9999 | >0.9999 |
| | Median (Q$_1$/Q$_3$) | 1 (0/1) | 1 (0/1) | 1 (0/1) | | | |
| | 95% CI | [0.5–0.94] | [0.51–1.09] | [0.35–0.79] | | | |
| | Abnormal (>3.09) | 0 | 0 | 0 | | | |
| **Ulnar (Upt)** | n | 53 | 49 | 30 | | | |
| | Mean ± SD | 0.49 ± 0.58 | 0.61 ± 0.76 | 0.47 ± 0.57 | >0.9999 | >0.9999 | >0.9999 |
| | Median (Q$_1$/Q$_3$) | 0 (0/1) | 1 (0/1) | 0 (0/1) | | | |
| | 95% CI | [0.33–0.65] | [0.39–0.83] | [0.25–0.68] | | | |
| | Abnormal (>2.23) | 0 | 1 (2.04%) | 0 | | | |
| **Ulnar (Ut)** | n | 53 | 49 | 30 | | | |
| | Mean ± SD | 0.72 ± 0.69 | 0.55 ± 0.68 | 0.57 ± 0.68 | 0.5250 | 0.8864 | >0.9999 |
| | Median (Q$_1$/Q$_3$) | 1 (0/1) | 0 (0/1) | 0 (0/1) | | | |
| | 95% CI | [0.53–0.91] | [0.36–0.75] | [0.31–0.82] | | | |
| | Abnormal (>2.79) | 0 | 0 | 0 | | | |
| **Common Fibular** | n | 53 | 48 | 30 | | | |
| | Mean ± SD | 0.87 ± 0.68 | 1.60 ± 3.21 | 0.33 ± 0.66 | >0.9999 | 0.0017* | 0.0029* |
| | Median (Q$_1$/Q$_3$) | 1 (0/1) | 1 (0/1,75) | 0 (0/0,25) | | | |
| | 95% CI | [0.68–1.06] | [0.67–2.54] | [0.09–0.58] | | | |
| | Abnormal (>2.91) | 0 | 4 (8.33%) | 0 | | | |
| **Tibial (Tpt)** | n | 53 | 49 | 30 | | | |
| | Mean ± SD | 0.83 ± 0.64 | 1.16 ± 1.20 | 0.47 ± 0.51 | >0.9999 | 0.0665 | 0.0168* |
| | Median (Q$_1$/Q$_3$) | 1 (0/1) | 1 (0/2) | 0 (0/1) | | | |
| | 95% CI | [0.65–1.01] | [0.83–1.54] | [0.28–0.66] | | | |
| | Abnormal (>2.75) | 0 | 1 (2.04%) | 0 | | | |
| **Tibial (Tt)** | n | 53 | 49 | 30 | | | |
| | Mean ± SD | 0.94 ± 0.79 | 1.02 ± 0.95 | 0.73 ± 0.74 | >0.9999 | 0.7977 | 0.5653 |
| | Median (Q$_1$/Q$_3$) | 1 (0/1,5) | 1 (0/1,5) | 1 (0/1) | | | |
| | 95% CI | [0.72–1.16] | [0.77–1.35] | [0.46–1.01] | | | |
| | Abnormal (>3.31) | 0 | 0 | 0 | | | |

HC = household contacts; n = number of nerves; SD = standard deviation; Q$_1$ = 25th percentile; Q$_3$ = 75th percentile; 95% CI– 95% confidence interval; Abnormal = number of nerves with abnormal measurements and percentages in parentheses; *p1 value* = healthy volunteers vs. seropositive HC; *p2 value* = healthy volunteers vs. seronegative HC; *p3 value* = seropositive HC vs. seronegative HC.

*Statistically significant.

We observed that neural enlargement detected by high-resolution US in SPHC may precede the classic clinical symptoms of leprosy and indicate that subclinical neuropathy may be the first manifestation of leprosy [13,23]. Furthermore, previous studies have demonstrated that leprosy patients may show abnormal nerve anatomy with preserved nerve function and vice versa [19,38]. Therefore, there is growing interest in the idea that US should be performed

**Table 5. Absolute difference in cross-sectional area (CSA) measurements (mm$^2$) between tunnel nerves and those proximal to the tunnel points on the same side by high-resolution ultrasonography in healthy volunteers and leprosy household contacts.**

| Nerves | | Healthy volunteers | Seropositive HC | Seronegative HC | p1 value | p2 value | p3 value |
|---|---|---|---|---|---|---|---|
| **Median (ΔMtpt)** | n | 106 | 90 | 57 | | | |
| | Mean ± SD | 1.71 ± 1.25 | 1.61 ± 1.04 | 1.40 ± 0.82 | >0.9999 | >0.9999 | >0.9999 |
| | Median (Q$_1$/Q$_3$) | 1 (1/2) | 2 (1/2) | 2 (1/2) | | | |
| | 95% CI | [1.47–1.95] | [1.4–1.85] | [1.25–1.66] | | | |
| | Abnormal (>5.46) | 0 | 0 | 0 | | | |
| **Ulnar (ΔUtpt)** | n | 106 | 98 | 60 | | | |
| | Mean ± SD | 1.09 ± 0.88 | 1.07 ± 0.91 | 0.98 ± 0.72 | >0.9999 | >0.9999 | 0.7345 |
| | Median (Q$_1$/Q$_3$) | 1 (0/2) | 1 (0/2) | 1 (0,25/1) | | | |
| | 95% CI | [0.93–1.26] | [0.89–1.25] | [0.79–1.17] | | | |
| | Abnormal (>3.73) | 0 | 1 (1.0%) | 0 | | | |
| **Tibial (ΔTtpt)** | n | 106 | 98 | 60 | | | |
| | Mean ± SD | 0.8 ± 0.60 | 0.60 ± 0.71 | 0.47 ± 0.60 | >0.9999 | 0.8095 | 0.8005 |
| | Median (Q$_1$/Q$_3$) | 1 (0/1) | 0 (0/1) | 0 (0/1) | | | |
| | 95% CI | [0.47–0.7] | [0.46–0.76] | [0.31–0.62] | | | |
| | Abnormal (>2.38) | 0 | 0 | 0 | | | |

HC = household contacts; n = number of nerves; SD = standard deviation; Q$_1$ = 25th percentile; Q$_3$ = 75th percentile; 95% CI– 95% confidence interval; *p1 value* = healthy volunteers vs. seropositive HC; *p2 value* = healthy volunteers vs. seronegative HC; *p3 value* = seropositive HC vs. seronegative HC.
*Statistically significant.

in addition to the neurophysiological study during the investigation of peripheral neuropathies [19,29,38–40].

In the present research, as previously observed in leprosy patients [17–22,30], anatomical abnormalities in peripheral nerves of HC were also found, reinforcing US as a useful tool for the diagnosis of early neural impairment in leprosy [23]. Nerve palpation is considered a subjective and low-accuracy method [17,19,22,37]. Furthermore, US evaluation may detect a greater extent of nerve thickening and a greater number of affected nerves when compared with clinical assessment [16,18,39–40].

In our study, the common fibular nerve was the most commonly involved nerve in SPHC, presenting significantly higher CSA values, in agreement with previous studies reporting that this nerve can be affected even in the early course of the disease [13,29]. We also observed significantly greater CSA values of tibial nerves in this group, a less commonly evaluated nerve by US, even among leprosy patients [21,29]. Thus, our findings suggest the importance of US

**Table 6. Analysis of dependence relation between peripheral neural enlargement demonstrated by high-resolution ultrasonography and the variables ELISA anti-PGL-I, slit-skin smear qPCR, skin biopsy qPCR and BCG scar, through multiple logistic regression.**

| | Simple Logistic Regression | | | Multiple Logistic Regression | | | |
|---|---|---|---|---|---|---|---|
| Predictor factors | *p* | ODDS | 95% CI | *p* | ODDS | 95% CI | Dependence Relation |
| ELISA anti-PGL-I | 0.0278* | 10.48 | (1.29–84.83) | 0.0311* | 10.48 | (1.24–88.61) | **Yes** |
| Slit-skin smear qPCR | 0.5727 | 1.64 | (0.29–9.12) | - | - | - | **No** |
| Skin biopsy qPCR | 0.6979 | 1.59 | (0.15–16.52) | - | - | - | **No** |
| BCG scar | 0.0110* | 0.19 | (0.05–0.68) | 0.0184* | 0.19 | (0.05–0.75) | **Yes** |

ELISA = enzyme-linked immunosorbent assay; anti-PGL-I = anti-phenolic glycolipid I; qPCR = Real Time Quantitative Polymerase Chain Reaction; BCG = Bacillus Calmette-Guérin.
*Statistically significant.

evaluation of lower-limb nerves in this population, not only of the common fibular nerve, but also of the tibial nerve [16,22].

We also found asymmetrical nerve impairment of the common fibular and tibial nerves proximal to the tunnel (Tpt) between right and left sides (ΔCSAs), detected by US evaluation in seropositive HC compared to seronegative ones, consistent with results of previous studies that stated asymmetric neural impairment as a classic pattern of leprosy neuropathy [14,20,29].

Our findings indicated that ELISA anti-PGL-I as the most important screening test for defining the increased chance of neural enlargement in leprosy contacts, in agreement with prior studies [7,13,25,26,41–43]. The use of ELISA anti-PGL-I test is attributable to its high correlation with multibacillary clinical forms [43]. Hence, this laboratory assay may help identify individuals with higher chances of developing leprosy neuropathy, not only justifying the treatment in those with confirmed diagnosis, but also indicating chemoprophylaxis for susceptible individuals. All contacts who showed neural thickening through the US were considered cases of primary neural leprosy and underwent specific treatment for leprosy [44].

Based on our results, the presence of one or more BCG scars provided protection against neural impairment, corroborating prior studies that showed an association between the vaccine and the prevention of leprosy, especially the multibacillary forms [41,45]. Therefore, in order to protect against leprosy, it is suggested to maintain the booster dose in leprosy control programs.

Although qPCR positivity in slit-skin smear and in skin biopsy did not determine a greater chance of neural thickening in our study, probably due to the small sample size, these tools are extremely useful for an early diagnosis and also to start the treatment in HC [12,26,28,44].

One limitation of our study is the difficulty of evaluating distal cutaneous branches by US, usually attributable to the initial sensory symptoms of leprosy neuropathy, as demonstrated before in several clinical studies [45–47]. Although this is one of the first studies to evaluate and detect nerve thickening in leprosy HC using high-resolution US, the number of subjects in each group was small and more studies need to be performed to confirm our findings, aiming to improve the efficacy of high-resolution US diagnosis in the identification of subclinical neuropathy.

Finally, the present study was innovative and proved to be useful in the early detection of neural thickening in leprosy SPHC. Therefore, our findings highlight the relevance of high-resolution US for evaluating peripheral nerves during the follow-up of leprosy HC with positive serology, strengthening the importance of epidemiological surveillance in this population.

## Supporting information

**S1 Table. CSA measurements of each leprosy household contact included in the study.** ID: Patient identification; Upt: Ulnar nerve proximal to the cubital tunnel; Ut: Ulnar nerve at the cubital tunnel; Mpt: Median nerve proximal to the carpal tunnel; Mt: Median nerve at the carpal tunnel; Tpt: Tibial nerve proximal to the tarsal tunnel; Tt: Tibial nerve at the tarsal tunnel; NI: Measurement not included (evidence of carpal tunnel syndrome; previous history of fibula fracture).
(DOCX)

**S2 Table. Clinical data of each leprosy household contact included in the study.** ID: Patient identification; ELISA: Enzyme-linked immunosorbent assay; anti-PGL-I: Anti-phenolic glycolipid I; ED: Extradomiciliary; ID: Intradomiciliary; PB: Paucibacillary; MB: Multibacillary; qPCR: Real Time Quantitative Polymerase Chain Reaction; BCG = Bacillus Calmette-Guérin.

US: Ultrasonography.
(DOCX)

**S3 Table. Epidemiological data and CSA measurements of each healthy volunteer included in the study.** ID: Participant identification; Upt: Ulnar nerve proximal to the cubital tunnel; Ut: Ulnar nerve at the cubital tunnel; Mpt: Median nerve proximal to the carpal tunnel; Mt: Median nerve at the carpal tunnel; Tpt: Tibial nerve proximal to the tarsal tunnel; Tt: Tibial nerve at the tarsal tunnel.
(DOCX)

# Acknowledgments

We are grateful for the contribution of the staff of the National Reference Center for Sanitary Dermatology and Leprosy (CREDESH) in carrying out this research, ensuring excellent care for participants involved in this study.

# Author Contributions

**Conceptualization:** Andrea De Martino Luppi, Diogo Fernandes dos Santos, Marcello Henrique Nogueira-Barbosa, Isabela Maria Bernardes Goulart.

**Data curation:** Andrea De Martino Luppi, Guilherme Emilio Ferreira, Denis Luiz Prudêncio, Douglas Eulálio Antunes, Lúcio Araújo, Isabela Maria Bernardes Goulart.

**Formal analysis:** Andrea De Martino Luppi, Guilherme Emilio Ferreira, Douglas Eulálio Antunes, Lúcio Araújo, Isabela Maria Bernardes Goulart.

**Funding acquisition:** Isabela Maria Bernardes Goulart.

**Investigation:** Andrea De Martino Luppi, Denis Luiz Prudêncio, Diogo Fernandes dos Santos, Marcello Henrique Nogueira-Barbosa, Isabela Maria Bernardes Goulart.

**Methodology:** Andrea De Martino Luppi, Denis Luiz Prudêncio, Douglas Eulálio Antunes, Lúcio Araújo, Diogo Fernandes dos Santos, Marcello Henrique Nogueira-Barbosa, Isabela Maria Bernardes Goulart.

**Project administration:** Isabela Maria Bernardes Goulart.

**Resources:** Isabela Maria Bernardes Goulart.

**Supervision:** Marcello Henrique Nogueira-Barbosa, Isabela Maria Bernardes Goulart.

**Validation:** Andrea De Martino Luppi, Guilherme Emilio Ferreira, Denis Luiz Prudêncio, Douglas Eulálio Antunes, Lúcio Araújo, Diogo Fernandes dos Santos, Marcello Henrique Nogueira-Barbosa, Isabela Maria Bernardes Goulart.

**Visualization:** Andrea De Martino Luppi, Guilherme Emilio Ferreira, Douglas Eulálio Antunes, Lúcio Araújo, Diogo Fernandes dos Santos, Marcello Henrique Nogueira-Barbosa, Isabela Maria Bernardes Goulart.

**Writing – original draft:** Andrea De Martino Luppi, Guilherme Emilio Ferreira.

**Writing – review & editing:** Andrea De Martino Luppi, Guilherme Emilio Ferreira, Douglas Eulálio Antunes, Diogo Fernandes dos Santos, Marcello Henrique Nogueira-Barbosa, Isabela Maria Bernardes Goulart.

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
