## [Decision Letter · Decision Letter 0]

19 Jan 2023

PONE-D-22-34737High-resolution ultrasonography for early diagnosis of neural impairment in seropositive leprosy household contactsPLOS ONE

Dear Dr. De Martino Luppi,

Thank you for submitting your manuscript to PLOS ONE. After careful consideration, we feel that it has merit but does not fully meet PLOS ONE’s publication criteria as it currently stands. Therefore, we invite you to submit a revised version of the manuscript that addresses the points raised during the review process. I believe that addressing the concerns of the reviewers could vastly improve the paper.

We look forward to receiving your revised manuscript.

Kind regards,

Nafis Faizi, MD, MPH

Academic Editor

PLOS ONE

Journal Requirements:

Reviewers' comments:

Reviewer's Responses to Questions

**Comments to the Author**

1. Is the manuscript technically sound, and do the data support the conclusions?

Reviewer #1: Yes

Reviewer #2: Partly

Reviewer #3: Yes

2. Has the statistical analysis been performed appropriately and rigorously? 

Reviewer #1: Yes

Reviewer #2: No

Reviewer #3: Yes

3. Have the authors made all data underlying the findings in their manuscript fully available?

Reviewer #1: Yes

Reviewer #2: No

Reviewer #3: Yes

4. Is the manuscript presented in an intelligible fashion and written in standard English?

Reviewer #1: No

Reviewer #2: Yes

Reviewer #3: Yes

5. Review Comments to the Author

Reviewer #1: Congratulations to the authors for this valuable work in the filed of leprosy which is one of the WHO neglected tropical diseases.

Detection of early signs of nerve damage can help identify subclinical cases which can then be targeted by chemoprophylaxis to break the chain of transmission.

The results of the study support the conclusions that HRUS along with serological testing can be employed to detect subclinical contacts.

The statistical analysis is rigorously performed. A suggestion would be to include how the sample size was calculated for this study.

The manuscript needs revision in terms of English language editing.

Some minor corrections/suggestions and clarifications:-

1. In the abstract, some abbreviations are used without defining like HHCs, HVs, US, CSA etc.

2. Under the heading of Introduction, ist sentence is too long. It can be rephrased and written in two parts.

3.Under "type of study and subjects", correction of tense is required; some sentences are in past tense, some in present.

4. Under "Clinical and laboratorial characterization of leprosy household contacts", ist two paragraphs seem redundant and can be removed. Also line 125 is not relevant to this section but to "Results" section.

5. The authors mention that "biopsies were performed on the small elbow skin fragment"; how was the biopsy sample taken?

6. In line 132, authors mention "Serum anti-PGL-1 IgM antibodies were detected by enzyme linked immunosorbent assay (ELISA) performed against the purified native PGL-I from the Mycobacterium leprae cell wall, according to a methodology previously described elsewhere." Where is this methodology described in the manuscript? Similarly with regards to PCR.

7. Lines 141-144 are not giving a clear meaning in the present form. They should be rephrased for clarity.

8. Rephrase the lines 180,181 as "considering any values greater than mean plus 3 standard deviations as abnormal" instead of "considering abnormal any values greater than the mean plus 3 standard deviations"

9.Under "Statistical analysis", Mann Whitney u test instead of Mann Whitney

10.In lines 288, 289, authors mention that "....a multivariate statistical method was conducted to confirm the dependence relation of variables...". It would be better to specify the statistical method. Also break up the sentence into two for clarity.

11. In the discussion, all the references are cited at the end of each paragraph instead of their relevant place in the paragraph. e.g,

the sentence " Nerve palpation is considered a subjective and low-accuracy method." should have a reference at the end of the sentence instead of paragraph end.

"In our study, the common fibular nerve was the most commonly involved nerve in seropositive HHCs, presenting significantly higher CSA values, in agreement with clinical studies reporting that this nerve can be affected even in the early course of the disease."; Give reference of the clinical studies referred here at the end of this sentence not grouped at the end of paragraph for better readability and reference.

"Our results indicated ELISA anti-PGL-1 as the most important screening test for defining the increased chance of neural enlargement in leprosy contacts, in agreement with prior studies." Reference for prior studies at the end of this sentence.

12. Rephrase the following sentence or break into two for clarity: "The US results, reported in this current research, have revealed anatomic changes in nerves also found in previous studies that investigated neuropathy using US in leprosy patients, reinforcing US as a useful tool for the diagnosis of peripheral nerve involvement in leprosy."

13. Last paragraph of discussion needs English language editing.

14. In lines 358, 359, authors mention "......follow-up of leprosy HHCs with recurrent positive serology over the

years....". Is recurrent positive serology a standard terminology? If yes, how is it defined? After how many positive serological tests, do you recommend US of peripheral nerves for early detection of nerve involvement or just having a positive serology is not enough criteria to test the nerves via US.

11. The authors mention that "In our study, we investigated all participants for the presence of Doppler signal.." Were other US parameters of nerves besides nerve thickness also studied like fascicular pattern etc.

12. Were skin biopsies subjected to histopathological examination or only for bacilli detection by PCR?

13. Was nerve biopsy done in cases who showed evidence of nerve involvement on US?

14. How were the seropositive cases who showed nerve involvement on US managed?

Reviewer #2: Comments: This manuscript reports a cross-sectional study to identify neural impairment in leprosy household contacts (HHC) by analyzing differences in high-resolution ultrasonographic measurements of peripheral nerves, considering their IgM anti-PGL-I serological status. They data were also compared to those of healthy volunteers.

This topic worth acknowledgment once we need new tools to help early diagnosis in leprosy. However, this manuscript needs some clarifications before publication.

Reviews and questions:

a) The authors should change PGL-1 to PGL-I in the whole manuscript, according to the original name and acronym of this glycoprotein (https://pubmed.ncbi.nlm.nih.gov/22439275/).

b) I suggest the authors to use the full text before using abbreviations or acronyms in the abstract.

c) The text in the lines 130-131 is not clear. I suggest reviewing it.

d) It is not clear if the healthy volunteers underwent the same full evaluation protocol, including clinical assessment to exclude leprosy.

e) It is important to inform that most HHC were also healthy subjects.

f) Include cm after number 4, line 165.

g) Lines 224-226 - It seems that something is missing in this sentence. In which group the participant were included? Was the CSA of this nerve to large?

h) Line 250 – IMPORTANT - As all ultrasound variables did not present normal probability distribution, the authors should use median and interquartile range instead of mean and standard deviation for the statistical analysis.

i) Line 295-296 and Table 6 – “The presence of at least one BCG vaccine scar demonstrated 5.2-fold greater protection against neural impairment (OR = 0.19; 95% CI: 0.05 – 0.75; p = 0.0184).”

Is that correct, considering an OR of 0.19?

The 95%CI values are different comparing the numbers on the paragraph with those in the table 6.

j) Line 307 – It is not the first study to investigate neural thickening in leprosy HHC through US assessment – See this very recent paper on the same topic https://www.frontiersin.org/articles/10.3389/fmed.2022.1059448/full

k) To help support argumentation in the lines 307-312 and 319-324, I suggest citing this reference: Voltan G, Filho FB, Leite MN, De Paula NA, Santana JM, Silva CML, Barreto JG, Da Silva MB, Conde G, Salgado CG and Frade MAC (2022) Point-of-care ultrasound of peripheral nerves in the diagnosis of Hansen’s disease neuropathy. Front. Med. 9:985252. doi: 10.3389/fmed.2022.985252

l) Lines 314-315 – “We observed that neural enlargement detected by high-resolution US in seropositive leprosy HHCs may precede the classic clinical symptoms of leprosy and indicate that subclinical neuropathy may be the first clinical manifestation of leprosy”. In my point of view, subclinical neuropathy cannot be the first “clinical” manifestation of leprosy once clinical assessment done by the physician was not able to detect any clinical manifestation.

m) Lines 350-352 - How was the sample size calculated? It is important to inform this in the methods section. The small sample size may have contributed to this finding. It constitutes a limitation to conclude that qPCR result has no correlation with US abnormalities.

n) Do the authors consider nerves abnormalities plus seropositivity as leprosy diagnosis? Anyway, it needs to be clarified in the discussion.

o) What US abnormalities means for those household contacts? Should they be treated with a full course of MDT, any kind of chemoprophylaxis or just followed up?

Reviewer #3: The authors performed a cross-sectional study composed of 3 groups, encompassing 49 seropositive Household Contacts (HHCs) 30 seronegative HHCs and 53 healthy volunteers who performed high-resolution ultrasonographic measurements of peripheral nerves to identify neural impairment in leprosy HHCs by analyzing differences in nerves between leprosy HHCs (seropositive and seronegative). They found a higher prevalence of neural thickening in leprosy-seropositive HHCs. The combination of positive anti-PGL-1 serology and absence of a BCG scar can identify individuals with greater chances of developing leprosy neuropathy, who should be referred for US examination, reinforcing the importance of including serological and imaging methods in the epidemiological surveillance of leprosy HHCs

This is an interesting finding that could identify de HHCs who should be followed closer. It can become an important tool for leprosy eradication efforts.

Comments:

1) On page 3 line 75, page 16 lines 305 and 308 and page 17 line 352, the authors describe that “there is no study available on the evaluation of peripheral nerve impairment by this technique encompassing leprosy 77 HHCs.

However there is a similar article which was published in 2023v January (Front. Med., 17 January 2023, Sec. Dermatology, Volume 9 2022,| https://doi.org/10.3389/fmed.2022.1059448; Silent peripheral neuropathy determined by high-resolution ultrasound among contacts of patients with Hansen's disease. That showed similar findings. In 83 Household Contacts of MB-patients that were submitted to peripheral nerve ultrasound and compared to 49 Health Volunteers and 176 Hansen Disease-patients.

2) On page 5 line 117 the authors describe that “contacts are evaluated for neurological impairment, through the presence of sensory symptoms or muscle weakness on clinical evaluation”. Why wasn't the sensory examination performed once the small fiber neuropathy characterized by impairment of thermal and pain sensory are the first neurological damage and at this time patients can not notice the sensory symptoms? Tt can be the first clinical sign of leprosy neuropathy.

3) Did other diseases were excluded from HHC individuals, such as carpal tunnel syndrome, hereditary neuropathy, Diabetes? It could influence the results obtained.

4) Is the nerve enlargement enough to define neuropathy? Another morphological parameter, such as fascicular echogenicity, is important to characterize neuropathy by the US.

5) The nerve size on USG can vary with age, sex, and size of the individual. The results of HHC and HV could have been paired.

6. PLOS authors have the option to publish the peer review history of their article (what does this mean?). If published, this will include your full peer review and any attached files.

Reviewer #1: No

Reviewer #2: No

Reviewer #3: **Yes: **Marcia Jardim

---

## [Author Response · Author response to Decision Letter 0]

13 Mar 2023

Uberlândia, 28th February, 2023

Ms. Ref. No.: PONE-D-22-34737 

Title: High-resolution ultrasonography for early diagnosis of neural impairment in seropositive leprosy household contacts.

Dear editor in chief,

We are very pleased to review and resubmit to PLOS ONE our manuscript entitled “High-resolution ultrasonography for early diagnosis of neural impairment in seropositive leprosy household contacts.” In the following paragraphs, we provide a point-by-point discussion on the comments made by the reviewers. The corrections are highlighted in red in the new version of the manuscript. We would like to thank the reviewers for the suggestions and to affirm that all the authors approved the final form of this reviewed manuscript. We sincerely and technically hope to clarify pending questions raised by our colleagues. 

Reviewers’ comments:

Reviewer #1:

Congratulations to the authors for this valuable work in the filed of leprosy which is one of the WHO neglected tropical diseases.

- Detection of early signs of nerve damage can help identify subclinical cases which can then be targeted by chemoprophylaxis to break the chain of transmission.

- The results of the study support the conclusions that HRUS along with serological testing can be employed to detect subclinical contacts.

- The statistical analysis is rigorously performed. A suggestion would be to include how the sample size was calculated for this study.

Response: 

The sample was performed for convenience: All individuals who met the inclusion criteria and underwent the proposed complementary tests were asked to be included in the study.

From September 2019 to March 2021, 805 household contacts were notified. A proportion of 70.1% (564/805) of these attended the initial evaluation, when all were submitted to anti-PGL-I serology collection. Among them, 22% (124/564) were seropositive. In this study, 39.5% (49/124) seropositive contacts were submitted to all complementary exams at the time when seropositivity to the ELISA anti-PGL-I IgM was confirmed. It is important to point out that, although we had a significant sample, the SARS-COV2 pandemic made it difficult for patients to access and adhere to it in a first assessment.

This information has been added to the text.

- The manuscript needs revision in terms of English language editing.

Response: We did a new English language editing.

Some minor corrections/suggestions and clarifications:-

1. In the abstract, some abbreviations are used without defining like HHCs, HVs, US, CSA etc. 

Response: These changes were made throughout the text.

2. Under the heading of Introduction, ist sentence is too long. It can be rephrased and written in two parts.

Response: We totally agree and corrected it in the text, as suggested.

3.Under "type of study and subjects", correction of tense is required; some sentences are in past tense, some in present. 

Response: We totally agree and corrected it in the text, as suggested.

4. Under "Clinical and laboratorial characterization of leprosy household contacts", ist two paragraphs seem redundant and can be removed. Also line 125 is not relevant to this section but to "Results" section.

Response: These changes were made throughout the text.

5. The authors mention that "biopsies were performed on the small elbow skin fragment"; how was the biopsy sample taken?

Response: This procedure was performed in all contacts and is now described in detail in the article's methodology.

6. In line 132, authors mention "Serum anti-PGL-I IgM antibodies were detected by enzyme linked immunosorbent assay (ELISA) performed against the purified native PGL-I from the Mycobacterium leprae cell wall, according to a methodology previously described elsewhere." Where is this methodology described in the manuscript? Similarly with regards to PCR.

Response: All laboratory descriptions were better described and checked by the authors.

7. Lines 141-144 are not giving a clear meaning in the present form. They should be rephrased for clarity.

Response: These changes were made throughout the text.

8. Rephrase the lines 180,181 as "considering any values greater than mean plus 3 standard deviations as abnormal" instead of "considering abnormal any values greater than the mean plus 3 standard deviations"

Response: This change was made throughout the text.

9.Under "Statistical analysis", Mann Whitney u test instead of Mann Whitney

Response: This change was made throughout the text.

10.In lines 288, 289, authors mention that "....a multivariate statistical method was conducted to confirm the dependence relation of variables...". It would be better to specify the statistical method. Also break up the sentence into two for clarity.

Response: Multiple logistic regression was utilized to verify the dependence relation between the presence of US abnormality (categorical variable) and the independent variables (ELISA anti-PGL-I IgM, slit-skin smear qPCR, skin biopsy qPCR and BCG scar), followed by the selection of variables by the stepwise method.

This information is described at the statistical analysis.

11. In the discussion, all the references are cited at the end of each paragraph instead of their relevant place in the paragraph. e.g,

the sentence " Nerve palpation is considered a subjective and low-accuracy method." should have a reference at the end of the sentence instead of paragraph end.

"In our study, the common fibular nerve was the most commonly involved nerve in seropositive HHCs, presenting significantly higher CSA values, in agreement with clinical studies reporting that this nerve can be affected even in the early course of the disease."; Give reference of the clinical studies referred here at the end of this sentence not grouped at the end of paragraph for better readability and reference.

"Our results indicated ELISA anti-PGL-1 as the most important screening test for defining the increased chance of neural enlargement in leprosy contacts, in agreement with prior studies." Reference for prior studies at the end of this sentence.

Response: We agree with the reviewer and have modified the way in which references are cited in the text.

12. Rephrase the following sentence or break into two for clarity: "The US results, reported in this current research, have revealed anatomic changes in nerves also found in previous studies that investigated neuropathy using US in leprosy patients, reinforcing US as a useful tool for the diagnosis of peripheral nerve involvement in leprosy."

Response: This sentence has been modified as suggested.

13. Last paragraph of discussion needs English language editing.

Response: This sentence has been modified as suggested.

14. In lines 358, 359, authors mention "......follow-up of leprosy HHCs with recurrent positive serology over the

years....". Is recurrent positive serology a standard terminology? If yes, how is it defined? After how many positive serological tests, do you recommend US of peripheral nerves for early detection of nerve involvement or just having a positive serology is not enough criteria to test the nerves via US. 

Response: This sentence was modified in the text, as only a positive serology already indicates the need for neural evaluation.

11. The authors mention that "In our study, we investigated all participants for the presence of Doppler signal.." Were other US parameters of nerves besides nerve thickness also studied like fascicular pattern etc.

Response: Other sonographic abnormalities can be considered in the morphological evaluation of the peripheral nerves. However, this study proposes a methodology that can be instituted as a public health policy. Therefore, we believe it more didactic to consider the presence of neural thickening as the main contact surveillance strategy.

12. Were skin biopsies subjected to histopathological examination or only for bacilli detection by PCR?

Response: One part of the skin sample was sent to the molecular pathology and biotechnology laboratory. The other part was sent to the institution’s pathology laboratory for histopathological evaluation. 

This information is very relevant. Therefore, it was added to the article.

13. Was nerve biopsy done in cases who showed evidence of nerve involvement on US?

Response: In order to perform a peripheral nerve biopsy, it is necessary to prove that there is an advanced degree of axonal involvement. In addition, it is only allowed to carry out nerve biopsy of purely sensitive branches, not evaluated by the method in question. Therefore, US is not a good tool to indicate nerve biopsy.

14. How were the seropositive cases who showed nerve involvement on US managed? 

Response: All contacts who showed neural thickening through the US were considered cases of primary neural leprosy and underwent specific treatment for leprosy. 

This information has been added to the text.

Reviewer #2:

Reviewer #2: Comments: This manuscript reports a cross-sectional study to identify neural impairment in leprosy household contacts (HHC) by analyzing differences in high-resolution ultrasonographic measurements of peripheral nerves, considering their IgM anti-PGL-I serological status. They data were also compared to those of healthy volunteers.

This topic worth acknowledgment once we need new tools to help early diagnosis in leprosy. However, this manuscript needs some clarifications before publication.

Reviews and questions:

a) The authors should change PGL-1 to PGL-I in the whole manuscript, according to the original name and acronym of this glycoprotein (https://pubmed.ncbi.nlm.nih.gov/22439275/).

Response: These changes were made throughout the text.

b) I suggest the authors to use the full text before using abbreviations or acronyms in the abstract.

Response: These changes were made throughout the text.

c) The text in the lines 130-131 is not clear. I suggest reviewing it.

Response: All changes requested above have been made.

d) It is not clear if the healthy volunteers underwent the same full evaluation protocol, including clinical assessment to exclude leprosy.

Response: The group of healthy volunteers was composed of participants of the same population from an endemic region for leprosy, but without a history of domiciliary contact with cases of leprosy. It is noteworthy that all healthy volunteers were submitted to dermato-neurological examination to rule out leprosy, even in the absence of epidemiological antecedents.

This information has been added to the text.

e) It is important to inform that most HHC were also healthy subjects. 

Response: It is important to highlight that the seronegative contacts were also healthy subjects and had no other plausible cause for peripheral neuropathy.

This information has been added to the text.

f) Include cm after number 4, line 165.

Response: This modification was made to the text.

g) Lines 224-226 - It seems that something is missing in this sentence. In which group the participant were included? Was the CSA of this nerve to large?

Response: We excluded one measurement of the common fibular nerve from a subject of the SPHC group due to a previous history of fibula fracture, which may be related to damage to the respective nerve in that location [33]. In addition, eleven measurements of the Mt nerve were excluded (eight measurements from SPHC and three from SNHC group) due to ultrasonographic evidence of carpal tunnel syndrome (measurement at Mt corresponded to more than twice the measurement of the same nerve at the Mpt). 

This information has been added to the text.

h) Line 250 – IMPORTANT - As all ultrasound variables did not present normal probability distribution, the authors should use median and interquartile range instead of mean and standard deviation for the statistical analysis. 

Response: We agree with the reviewer and this information has been added to the tables.

i) Line 295-296 and Table 6 – “The presence of at least one BCG vaccine scar demonstrated 5.2-fold greater protection against neural impairment (OR = 0.19; 95% CI: 0.05 – 0.75; p = 0.0184).” 

Is that correct, considering an OR of 0.19?

The 95%CI values are different comparing the numbers on the paragraph with those in the table 6.

Response: As the OR is less than 1.0, it is a protection factor. Therefore, the best way to find the magnitude of this protection factor is by calculating:

Protection factor = 1/OR = 1.0/0.19 = 5.2

This information has been added to the text.

j) Line 307 – It is not the first study to investigate neural thickening in leprosy HHC through US assessment – See this very recent paper on the same topic https://www.frontiersin.org/articles/10.3389/fmed.2022.1059448/full

Response: This sentence has been modified and the recent publication referenced in our article.

k) To help support argumentation in the lines 307-312 and 319-324, I suggest citing this reference: Voltan G, Filho FB, Leite MN, De Paula NA, Santana JM, Silva CML, Barreto JG, Da Silva MB, Conde G, Salgado CG and Frade MAC (2022) Point-of-care ultrasound of peripheral nerves in the diagnosis of Hansen’s disease neuropathy. Front. Med. 9:985252. doi: 10.3389/fmed.2022.985252

Response: The article was cited in the text.

l) Lines 314-315 – “We observed that neural enlargement detected by high-resolution US in seropositive leprosy HHCs may precede the classic clinical symptoms of leprosy and indicate that subclinical neuropathy may be the first clinical manifestation of leprosy”. In my point of view, subclinical neuropathy cannot be the first “clinical” manifestation of leprosy once clinical assessment done by the physician was not able to detect any clinical manifestation.

Response: Although asymptomatic, it may be the first manifestation of the disease. In any case, we agree with the reviewer and have modified the text.

m) Lines 350-352 - How was the sample size calculated? It is important to inform this in the methods section. The small sample size may have contributed to this finding. It constitutes a limitation to conclude that qPCR result has no correlation with US abnormalities. 

Response: During the study period, 805 leprosy contacts were notified. A proportion of 70.1% (564/805) of these attended the initial evaluation, when all were submitted to anti-PGL-I serology collection. Among them, 22% (124/564) were seropositive. In this study, 39.5% (49/124) SPHC were submitted to all complementary exams at the time when seropositivity to the ELISA anti-PGL-I IgM was confirmed. It is important to point out that, although we had a significant sample, the SARS-COV2 pandemic made it difficult for patients to access the health service and adhere to it in a first assessment. 

This information was added to the article.

n) Do the authors consider nerves abnormalities plus seropositivity as leprosy diagnosis? 

Response: All contacts who showed neural thickening through the US were considered cases of primary neural leprosy and underwent specific treatment for leprosy.

This information has been added to the text.

o) What US abnormalities means for those household contacts? Should they be treated with a full course of MDT, any kind of chemoprophylaxis or just followed up? 

Response: All contacts with US abnormalities were treated with standard leprosy schemes, considering that it was a primary neural leprosy. Chemoprophylaxis is not yet a public health policy in Brazil and has not yet been instituted. However, chemoprophylaxis should only be used in the absence of clinical disease and/or abnormalities in the peripheral nerve detected by neurophysiological and/or morphological tests such as US.

Reviewer #3:

Reviewer #3: The authors performed a cross-sectional study composed of 3 groups, encompassing 49 seropositive Household Contacts (HHCs) 30 seronegative HHCs and 53 healthy volunteers who performed high-resolution ultrasonographic measurements of peripheral nerves to identify neural impairment in leprosy HHCs by analyzing differences in nerves between leprosy HHCs (seropositive and seronegative). They found a higher prevalence of neural thickening in leprosy-seropositive HHCs. The combination of positive anti-PGL-1 serology and absence of a BCG scar can identify individuals with greater chances of developing leprosy neuropathy, who should be referred for US examination, reinforcing the importance of including serological and imaging methods in the epidemiological surveillance of leprosy HHCs

This is an interesting finding that could identify de HHCs who should be followed closer. It can become an important tool for leprosy eradication efforts.

Comments:

1) On page 3 line 75, page 16 lines 305 and 308 and page 17 line 352, the authors describe that “there is no study available on the evaluation of peripheral nerve impairment by this technique encompassing leprosy 77 HHCs.

However there is a similar article which was published in 2023v January (Front. Med., 17 January 2023, Sec. Dermatology, Volume 9 2022,| https://doi.org/10.3389/fmed.2022.1059448; Silent peripheral neuropathy determined by high-resolution ultrasound among contacts of patients with Hansen's disease. That showed similar findings. In 83 Household Contacts of MB-patients that were submitted to peripheral nerve ultrasound and compared to 49 Health Volunteers and 176 Hansen Disease-patients.

Response: The writing was modified and the published article became a reference for our work.

2) On page 5 line 117 the authors describe that “contacts are evaluated for neurological impairment, through the presence of sensory symptoms or muscle weakness on clinical evaluation”. Why wasn't the sensory examination performed once the small fiber neuropathy characterized by impairment of thermal and pain sensory are the first neurological damage and at this time patients can not notice the sensory symptoms? Tt can be the first clinical sign of leprosy neuropathy. 

Response: All contacts underwent a rigorous sensory evaluation, ruling out the impairment of all sensory modalities (pain, temperature, touch and vibration).

This information has been added to the text.

3) Did other diseases were excluded from HHC individuals, such as carpal tunnel syndrome, hereditary neuropathy, Diabetes? It could influence the results obtained.

Response: Yes. All diseases that can cause peripheral neuropathy were ruled out in all groups.

This information is described at the Type of study and subjects.

4) Is the nerve enlargement enough to define neuropathy? Another morphological parameter, such as fascicular echogenicity, is important to characterize neuropathy by the US. 

Response: Although the presence of neural thickening is not a pathognomonic alteration of leprosy, the presence of this alteration combined with family epidemiological antecedents allows us to conclude the diagnosis of leprosy. Although other US parameters can be used, the article proposes a method that is easier to perform, such as calculating the cross-sectional area. 

5) The nerve size on USG can vary with age, sex, and size of the individual. The results of HHC and HV could have been paired. 

Response: This is a cross-sectional study composed of 3 groups, encompassing 49 seropositive HC (SPHC), 30 seronegative HC (SNHC) and 53 healthy volunteers (HV), enrolled by intentional sampling from September 2019 to March 2021. From this period, 805 HC were notified. A proportion of 70.1% (564/805) of these attended the initial evaluation, when all were submitted to anti-PGL-I serology collection. Among them, 22% (124/564) were seropositive. In this study, 39.5% (49/124) SPHC were submitted to all complementary exams at the time when seropositivity to the ELISA anti-PGL-I IgM was confirmed. It is important to point out that although we had a significant sample, the SARS-COV2 pandemic made it difficult for patients to access the health service and adhere to it in a first assessment. Although we did not carry out this pairing in relation to sex and age, we used a safe margin of 3 standard deviations in relation to the means obtained. Furthermore, our data are compatible with the values used in other studies. This information has been added to the text.

We are looking forward to hearing from you soon. 

Sincerely,

Isabela Maria Bernardes Goulart, MD, PhD.

Corresponding Author: imbgoulart@gmail.com (IMBG)

---

## [Decision Letter · Decision Letter 1]

24 Apr 2023

High-resolution ultrasonography for early diagnosis of neural impairment in seropositive leprosy household contacts

PONE-D-22-34737R1

Dear Dr. De Martino Luppi,

We’re pleased to inform you that your manuscript has been judged scientifically suitable for publication and will be formally accepted for publication once it meets all outstanding technical requirements.

Kind regards,

Nafis Faizi, MD, MPH

Academic Editor

PLOS ONE

Additional Editor Comments (optional):

Reviewers' comments:

Reviewer's Responses to Questions

**Comments to the Author**

1. If the authors have adequately addressed your comments raised in a previous round of review and you feel that this manuscript is now acceptable for publication, you may indicate that here to bypass the “Comments to the Author” section, enter your conflict of interest statement in the “Confidential to Editor” section, and submit your "Accept" recommendation.

Reviewer #1: All comments have been addressed

Reviewer #3: All comments have been addressed

2. Is the manuscript technically sound, and do the data support the conclusions?

Reviewer #1: Yes

Reviewer #3: Yes

3. Has the statistical analysis been performed appropriately and rigorously? 

Reviewer #1: I Don't Know

Reviewer #3: Yes

4. Have the authors made all data underlying the findings in their manuscript fully available?

Reviewer #1: Yes

Reviewer #3: Yes

5. Is the manuscript presented in an intelligible fashion and written in standard English?

Reviewer #1: Yes

Reviewer #3: Yes

6. Review Comments to the Author

Reviewer #1: Most of the suggestions have been incorporated.

It was noted that no data on the pathological findings has been provided as is mentioned in the methodology that the skin biopsy sample was subjected to PCR and histopathological examination. It will be pertinent to mention the pathology findings especially in case of SPHC.

Reviewer #3: Congratulations for this interesting work which can become an important tool for leprosy eradication efforts.

7. PLOS authors have the option to publish the peer review history of their article (what does this mean?). If published, this will include your full peer review and any attached files.

Reviewer #1: No

Reviewer #3: **Yes: **Marcia Rodrigues Jardim

---

## [Editor Report · Acceptance letter]

10 May 2023

PONE-D-22-34737R1 

High-resolution ultrasonography for early diagnosis of neural impairment in seropositive leprosy household contacts 

Dear Dr. Goulart:

I'm pleased to inform you that your manuscript has been deemed suitable for publication in PLOS ONE. Congratulations! Your manuscript is now with our production department. 

Kind regards, 

on behalf of

Dr. Nafis Faizi 

Academic Editor

PLOS ONE